# Defining Wildfire Susceptibility Maps in Italy for Understanding Seasonal Wildfire Regimes at the National Level

Andrea Trucchia [1,*], Giorgio Meschi [1], Paolo Fiorucci [1], Andrea Gollini [2] and Dario Negro [2]

1 CIMA Research Foundation, I-17100 Savona, Italy; giorgio.meschi@cimafoundation.org (G.M.); paolo.fiorucci@cimafoundation.org (P.F.)
2 Italian Department of Civil Protection, Presidency of the Council of Ministers, I-00189 Rome, Italy; Andrea.Gollini@protezionecivile.it (A.G.); Dario.Negro@protezionecivile.it (D.N.)
* Correspondence: andrea.trucchia@cimafoundation.org

**Abstract:** Wildfires constitute an extremely serious social and environmental issue in the Mediterranean region, with impacts on human lives, infrastructures and ecosystems. It is therefore important to produce susceptibility maps for wildfire management. The wildfire susceptibility is defined as a static probability of experiencing wildfire in a certain area, depending on the intrinsic characteristics of the territory. In this work, a machine learning model based on the Random Forest Classifier algorithm is employed to obtain national scale susceptibility maps for Italy at a 500 m spatial resolution. In particular, two maps are produced, one for each specific wildfire season, the winter and the summer one. Developing such analysis at the national scale allows for having a deep understanding on the wildfire regimes furnishing a tool for wildfire risk management. The selected machine learning model is capable of associating a data-set of geographic, climatic, and anthropic information to the synoptic past burned area. The model is then used to classify each pixel of the study area, producing the susceptibility map. Several stages of validation are proposed, with the analysis of ground retrieved wildfire databases and with recent wildfire events obtained through remote sensing techniques.

**Keywords:** wildfire susceptibility mapping; machine learning; wildfire management

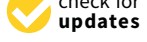



## 1. Introduction

Wildfires represent a hazardous and harmful phenomenon to people and the environment, especially in populated areas. Among the weather-induced emergencies, they constitute one of the most complex scenarios, since wildfires constitute a nonlinear, multiscale and multi-physics process. According to the EFFIS Annual Report on Forest Fires in Europe, Middle East and North Africa [1], in the European Union, over 3400 km² of land were burned in 2020. Around 40% of the burned area was part of the EU's Natura 2000 network [2], that is, the European coordinated network of protected areas, focused on ensuring the long-term survival of Europe's most valuable and threatened habitats and species. The damage caused in many of these ecosystems will likely take many years to be restored. Moreover, the summer wildfire season of 2021 ravaged Mediterranean countries, with more than half a million burnt hectares, and took a high toll on firefighter and civilian lives.

Climate change effects on wildfire regimes are becoming more noticeable year after year too. According to the EFFIS report [1], there is a clear trend showing increasing levels of fire danger, longer fire seasons, and more frequent fast spreading 'mega fires', over which the traditional firefighting strategies have little power. In general, fires no longer affect Southern European states only, but they constitute a growing threat also for central and northern Europe.

However, it should also be noted that, if a larger temporal time window is analyzed, a decrease in burned area and number of fires is observed in Italy and most Southern European countries, as demonstrated by Turco et al. [3]. Such negative trends at the large

time scale can be explained by an increased effort in fire management and prevention after the big wildfire seasons of the 1980s. According to Turco et al., a local increase in wildfire activity may be related to recent socioeconomic transformations, which may lead to more dangerous landscape configurations and also to climate change effects. Currently, fire management is focused mainly on fire suppression, which can lead to higher fuel load and fuel connectivity [3,4]. The next fire management strategies should thus improve prevention and adaptation measures, in addition to the sole suppression. The use of wildfire static and dynamic models, such as the seasonal susceptibility maps proposed in this work, helps in landscape management, prevention, and land use planning, during this overall decrease in fire activity, and gives to Civil Protection Authorities and decision makers the correct tools to tackle occasional outliers in wildfire seasonal trends.

Speaking of the causes, more than nine out of ten fires in the EU are human-caused [1,5]. In addition, Italy is characterized by human-caused fires. Only two percent of fires are due to natural causes (lightning) while the other part is due to human activities. The latter can be divided in intentional fires due to renewal pastures and acts of arson, and unintentional fires due to equipment use and malfunctions, negligently discarded cigarettes and burning plant debris during agriculture and forestry activities [1,5]. Italy is characterized by an increasing wildfire activity during recent years. In particular, in 2020, more wildfires were recorded than in 2019. The number of fires and the burned area increased respectively by 12% and 38%. This increase is mainly related to the severe wildfire activity in Sicily and Sardinia regions. This trend has then been validated by the catastrophic outcomes of the 2021 summer season of wildfires, where the fire events of Sardinia, Sicily and Calabria made the latter season even more severe in terms of burned hectares than the *annus horribilis* of 2017.

Socio-demographic changes in rural areas, such as the abandonment of agricultural lands in the crops-forest interface combined with climate change effects, have created unprecedented and challenging circumstances, which call for an improvement in methodologies and tools for wildfire management and fire reduction. Susceptibility maps represent a valid tool for wildfire management activities.

Despite its widespread use, the concept of wildfire susceptibility has been defined differently by several authors [6]. For example, Leuenberger et al. [7] defined wildfire susceptibility as "the probability that fire occurs in a specific area without considering a temporal scale, assessed on the basis of predisposing factors related to terrain's intrinsic characteristics," while in other works, such as in the one of *Cao et al.* [8], the wildfire susceptibility is defined as the spatially distributed "likelihood of suffering harm," thus allowing for factors that are not related to the terrain's intrinsic characteristics (e.g., the simple probability of wildfire occurence, retrieved from wildfire historical databases). In this work, the definition of Leuenberger et al. is adopted. The authors nonetheless stress the fact that the "probability" term used in the definition, however, widely used in a large number of works [9], should not be intended in the strictly rigorous mathematical sense, considering the susceptibility as an indicator, which ranges from 0 to 1, useful for discriminating the areas that are more fire-prone, giving priorities and useful information to several stages of wildfire management [10].

This quantitative evaluation is carried out by taking into account two aspects: the location and spatial extent of past wildfire occurrences, and the climatic, geo-environmental and anthropogenic predisposing factors (features) that are likely to be connected with wildfire spread. Among these features, the main chosen geo-environmental predisposing factors are the terrain elevation, slope, aspect (northing and easting) and vegetation cover while the historical means of temperature and cumulative precipitation account for climate features. Finally, anthropogenic factors like the distance from urban areas, roads, and cultivated land have been taken in consideration as they could give a hint on the human presence and anthropic activities, which could be related to a possible fire ignition. The definition itself of susceptibility maps relies on the assumption that future occurrences of wildfires are expected to take place under anthropic, climatic, and geo-environmental con-

ditions similar to the already occurred ones. Providing accurate and reliable susceptibility map is the first step when the objective is to develop an accurate risk mapping for wildfire risk management, with the potential wildfire intensity assessment and the identification of exposed assets and their vulnerabilities being the next natural steps.

Several approaches about wildfire susceptibility assessment can be found in literature. Several simple yet robust statistical models can be employed, see, e.g., [11–13]. Another approach involves the iteration of wildfire spread model runs with random ignition points and weather conditions, see, e.g., [14,15]. Recent advances in machine learning (ML) algorithms have attracted interest from the scientific community in susceptibility mapping for environmental problems [16–18]. Such approaches have been pursued in recent times to produce wildfire susceptibility maps in different world regions [19–24]. ML techniques are capable of learning from data, modeling the hidden relationships between input variables (features) and output (labels). In Ref. [25], a stochastic ML approach based on Random Forest (RF) elaborated wildfire susceptibility mapping for the Liguria region, Italy. The same approach, with some slight change, is adopted in this paper. In this work, a wildfire susceptibility mapping is performed in Italy on the national scale, based on an ML (RF) model. The ML model links the topographic, anthropic and climatic characteristics of the zones that experienced wildfires, producing season-specific maps. In the transition from regional to national mapping, it may be necessary to refer to more general global or European level data-sets, circumventing the problem of harmonizing local detailed information available at the regional scale [26]. For this reason, a coarser classification of the fuel map is adopted with respect to the one used in the pilot wildfire susceptibility study in Liguria [25]. In this experiment at the national scale, climatic variables, such as mean annual temperature and precipitation are considered, while the use of land that retrieved burned polygons is maintained in both approaches. This study demonstrates the ability of the RF technique to discern the most susceptible areas, even with national scale data, with the adopted set of predisposing factors.

## 2. Study Area

Italy is particularly affected by wildfires for its remarkable heterogeneity in topography and vegetation cover, its population density, and climatic conditions [27,28]. Blasi et al. [29–31] identified and mapped two divisions of the Italian territory, the Temperate and the Mediterranean one. The Temperate Division includes the Alps, the Po Plain, and most of the Apennines. It accounts for 64 percent of the national territory. This area is characterized by a general lack of summer aridity (less than two months) and by marked differences between summer and winter temperatures. The natural vegetation mainly consists of forests, with broad-leaved deciduous plants (*Quercus, Fagus* and *Carpinus* species). The Mediterranean Division includes the southern Apennines, the Tyrrhenian and Ionian coasts, the southern Adriatic coast, and the Islands; it accounts for almost 36 percent of the Italian territory. This area is characterized by summer aridity, with precipitations concentrated in autumn and winter. The natural vegetation mainly consists of mixed woods of evergreen and deciduous species, shrublands, and Mediterranean maquis. A representation of the CORINE 2018 land cover [32] classes corresponding to vegetated areas can be found in Figure 1. The label for each code is reported in the Supplementary Materials.

Italy has a total area of 301,340 km$^2$, about 20 percent of which is covered by protected areas of Europe's Natura 2000 network. These areas include the Sites of Community Importance (SCIs) and the Special Areas of Conservation (SACs), where the natural habitat is protected (Figure 2).

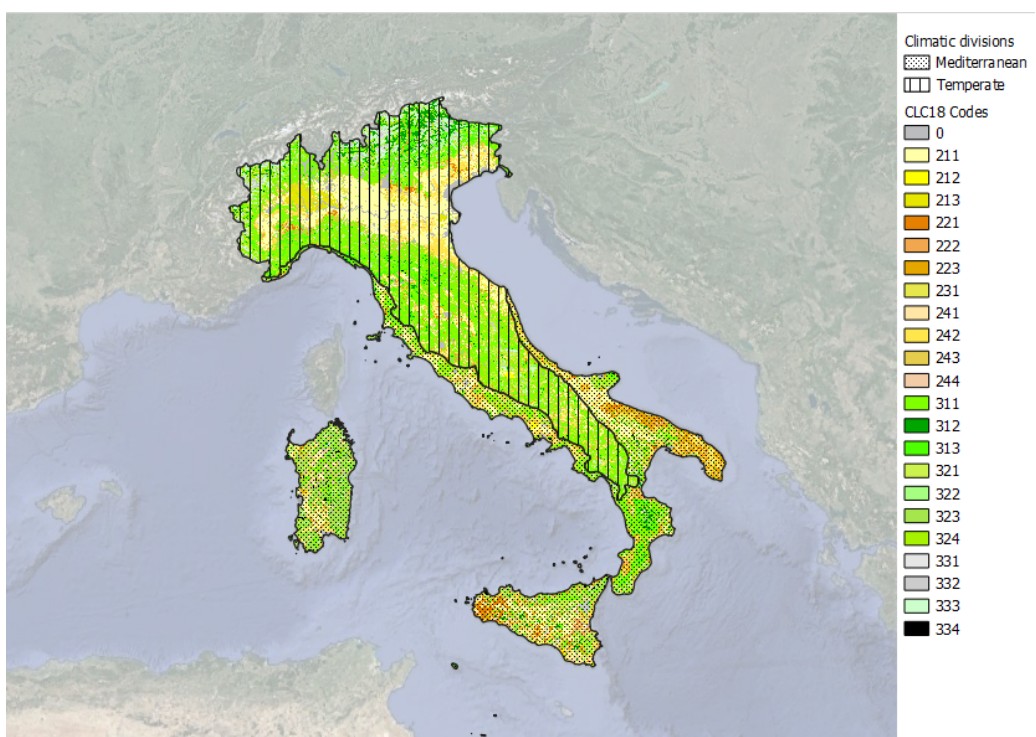

**Figure 1.** The distribution of the CORINE 2018 land cover classes in Italy. All non-burnable classes have been merged in the type 0. The two climatic subdivisions of Italy [30,31], the Mediterranean (dotted part) and the Temperate (hashed part) one, are also represented.

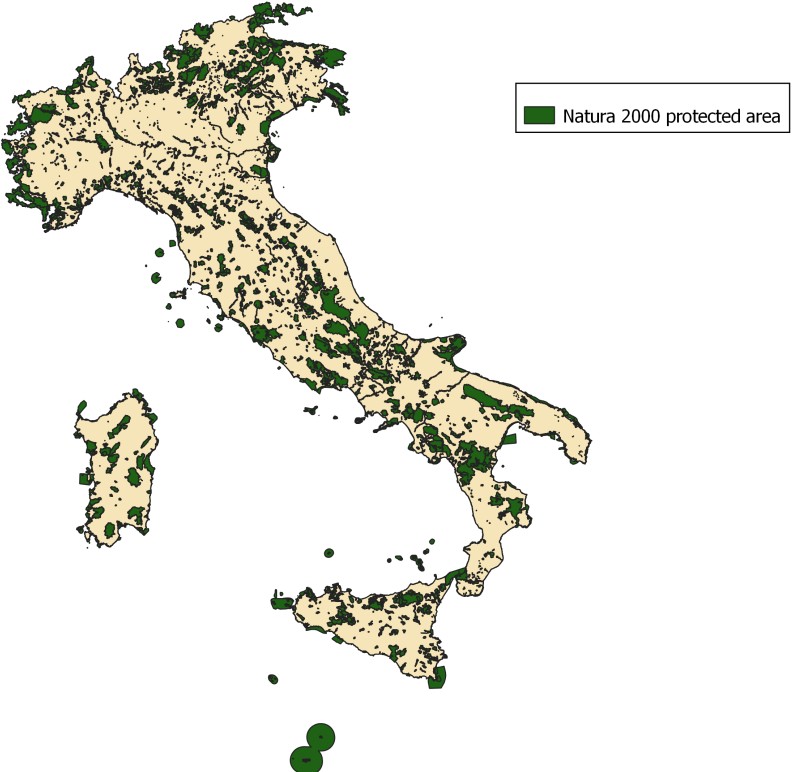

**Figure 2.** Spatial distribution of Natura 2000 protected areas in Italy.

Italy's varied geological structure contributes to its high climate and habitat diversity. The Italian peninsula is located at the center of the Mediterranean Sea, forming a corridor between central Europe and North Africa, with a total of 8000 km of coastline. Because

of the length of the peninsula and the mostly mountainous hinterland, the climate of Italy is highly diverse. In most of the inland northern and central regions, the climate ranges from humid subtropical to humid continental and oceanic. In particular, the climate of the Po river valley is mostly continental, with harsh winters and hot summers. The coastal areas of Liguria, Tuscany and most of the South are generally characterized by the Mediterranean climate.

## 3. Materials and Methods

In this section, the data used in the ML analysis are described in detail, as well as the adopted algorithms.

### 3.1. Historical Wildfire Database

A proper wildfire susceptibility mapping process relies on a data-set of past burned area polygons. Since 2000, Italy has started building a complete data-set of ground-retrieved burned area polygons, gathered at the regional level. Those polygons are collected on the ground and individually verified. They were acquired and disseminated by the Forestry Corps (*Corpo Forestale dello Stato*) until 2016 and by the *Carabinieri Forestali* since 2017, following the art 7 paragraph 2 of Legislative Decree 177/2016. The sources of this data-set are thus the *Arma dei Carabinieri*, the Italian Regions (*Regioni a Statuto ordinario e speciale*), and the Italian Autonomous Provinces. For the presented analysis, the past burned areas in the time window 2007–2019 have been considered.

Italy is characterized by two different wildfire regimes [25] corresponding to two macro seasons: the summer regime (from May to October) and the winter one (from November to April).

A representation of the burned area based on the seasonal division is presented in Figure 3. The yearly trends for the seasonal wildfires are available in Figures 4 and 5.

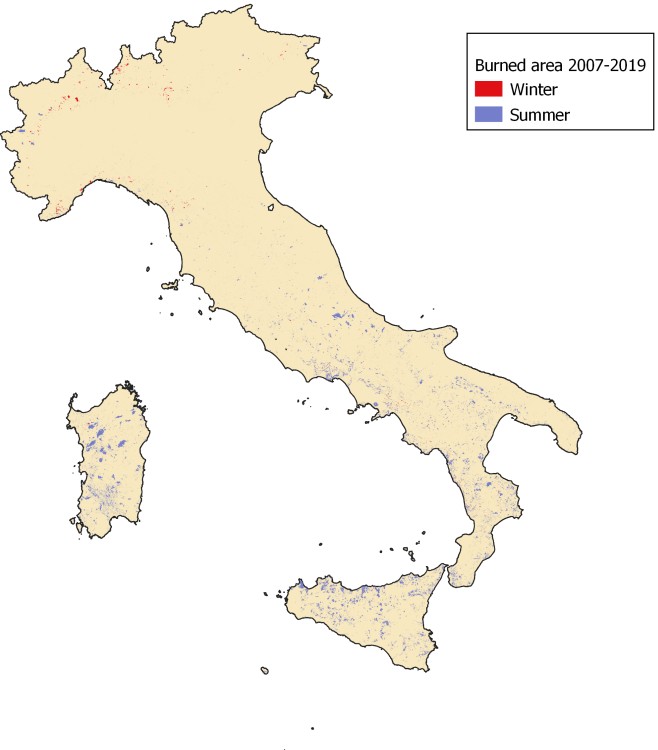

**Figure 3.** Spatial distribution of wildfire burned area in Italy for 2007 to 2019.

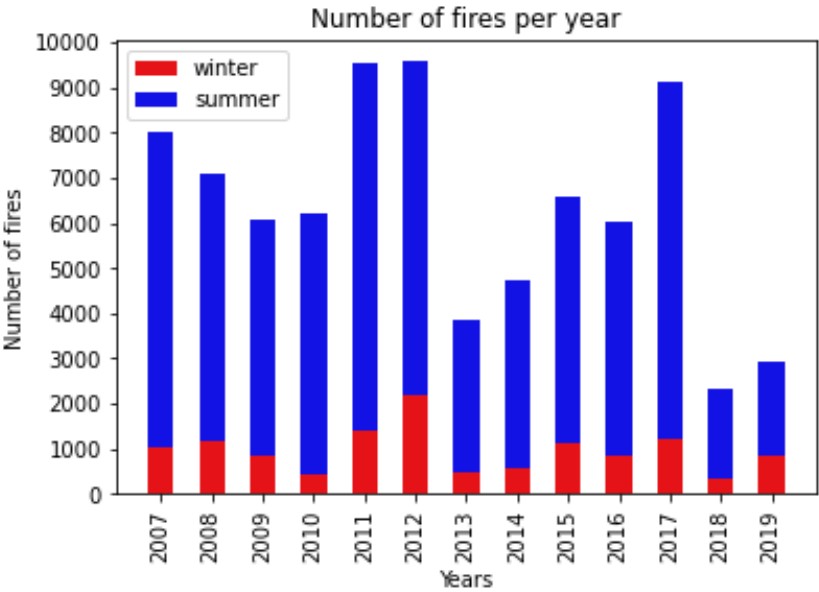

**Figure 4.** Seasonal number of fires in Italy per year based on the historical dataset used in this study (2007–2019). The wildfires occurred in winter season are presented in red while the ones occurred in summer season are presented in blue.

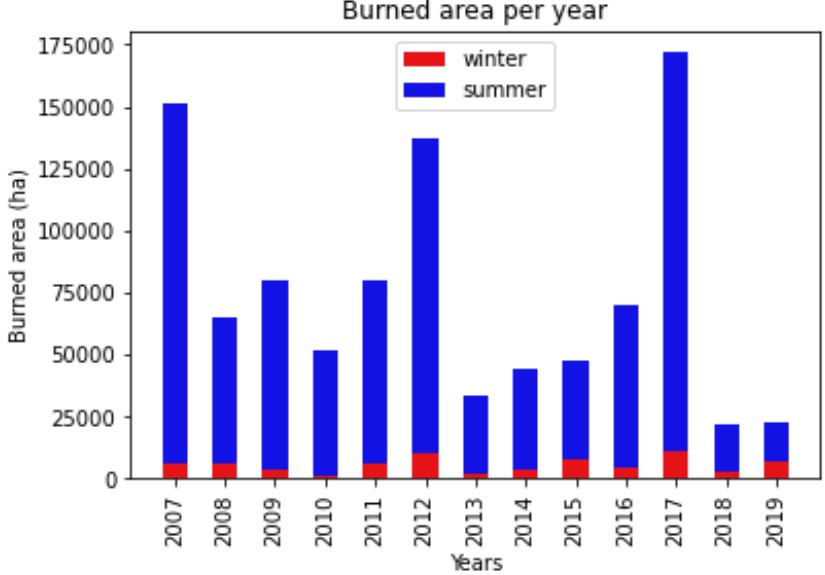

**Figure 5.** Seasonal wildfire burned area in Italy per year based on the historical dataset used in this study (2007–2019). The wildfires occurred in winter season are presented in red colour while the ones occurred in summer season are presented in blue colour.

The wildfire database has been preprocessed via *GIS* operations described in the following, to define the dependent variable in the ML modeling approach. This has been achieved by merging all the polygons corresponding to winter and summer wildfire seasons, respectively, and then discretizing them to the working raster, with projection *EPSG:32632 UTM zone 32N* and spatial resolution of 500 m. For both wildfire seasons, every pixel can be associated with a binary value, corresponding to the *label*, or dependent variable, of the classification process performed by the ML model: one if that pixel has experienced at least a fire according to the 12-year wildfire data-set, and zero if no fire occurrence has been recorded for the examined pixel.

It is remarked that, in this work, the information about the frequency of the wildfires' occurrence in a determined pixel is canceled, in order not to give too much importance to agricultural and pastures-based wildfires. This solution thus helps with reducing the impact of controlled annual fires on agricultural land that could lead to areas of high wildfire susceptibility that are not related to topographical causes but to specific man-made activities.

### 3.2. Additional Wildfire Data-Set for Model Validation

For validation purposes, another set of wildfires is considered in the presented work. This validation data-set consists of 55 wildfires located in Southern Italy (and Italian islands of Sicily and Sardinia), whose size ranges from 504 to 11,500 hectares; see Figure 6. All the analyzed events took place from June 2021 to the end of August 2021. Those quite recent wildfires have been detected by making use of Remote Sensing algorithms. In particular, Sentinel-2 multispectral data (20 m resolution) are elaborated via a fully automated processing chain for near real-time mapping of burned areas, AUTOBAM (AUTOmatic Burned Areas Mapper) [33,34]. AUTOBAM is configured to work daily on a national scale for the Italian territory to support the Italian Civil Protection Department in wildfire management issues. Its processing chain involves a Sentinel-2 data procurement component, an image processing algorithm, based on the relativized form of the delta normalized burn ratio and the normalized difference vegetation index, and the final delivery of the burned area maps to the end-user. At any rate, it is necessary to remark that such fires, which happened just months ago, are not still consolidated in the full fledged official database described in Section 3.1. However, they still constitute an interesting test for the adopted methodology, as shown in the next sections.

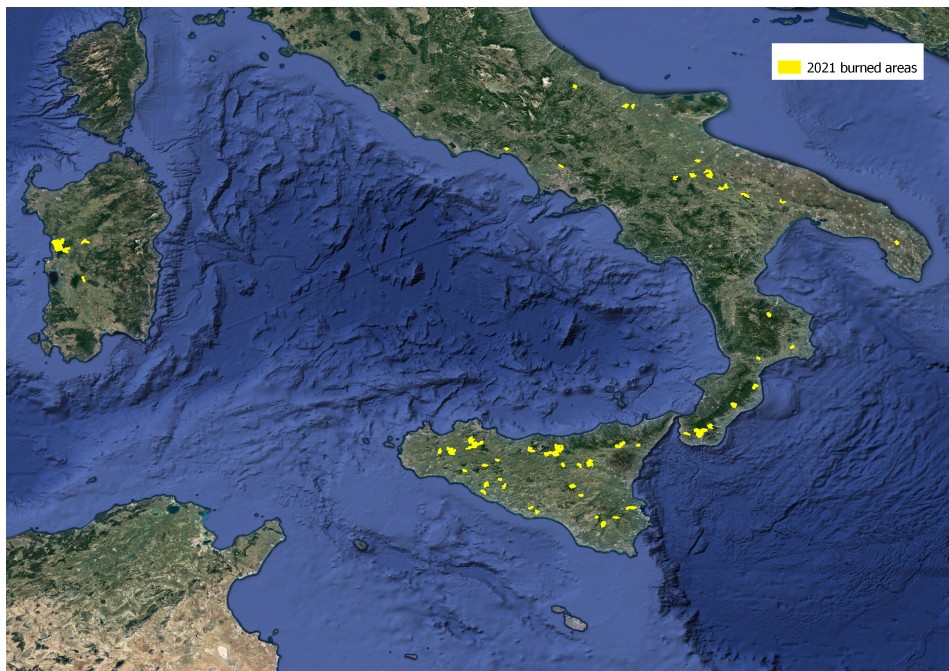

**Figure 6.** In this figure, the remote-sensing retrieved burned areas corresponding to 55 wild-fires greater than 500 hectares considered during the validation of the Summer susceptibility map are represented.

### 3.3. The Predisposing Factors: Geo-Climatic and Anthropic Data-Set

The predisposing factors are the explanatory variables of the ML modeling framework. In particular, a set of geo-climatic and anthropic variables has been defined with the aim of creating a data-set that associates with every spatial pixel (500 × 500 m) the information

related to the topography, the climate, and other factors related to the human presence. The explanatory variables are listed in Table 1.

**Table 1.** Input data of the machine learning model.

| Variable Name | Variable Type | No. of Variables |
| --- | --- | --- |
| DEM | Numerical (meters) | 1 |
| Slope | Numerical (degree) | 1 |
| Northness and Eastness | Numerical | 2 |
| Distance to anthropogenic features | Numerical (meters) | 4 |
| Area Natura 2000 | Binary | 1 |
| Vegetation type | Categorical (24 classes) | 1 |
| Mean Temperature, Mean Precipitation | Numerical(°C and mm) | 2 |
| Neighboring vegetation | Numerical (percentage) | 24 |

Topographic Variables

The 20-m resolution Digital Elevation Model (DEM) made available by ISPRA, the Italian Institute for Environmental Protection and Research, has been employed (http://www.sinanet.isprambiente.it/it/sia-ispra/download-mais/dem20/view) (accessed on 30 June 2021). Through GIS elaborations, the slope and aspect vector components Northing and Easting [35] have then been computed, and the results have been projected to the working resolution.

### 3.4. Climatic Variables

The climatic raster data for Italy have been retrieved from the ISPRA database—in particular, the gridded data from the GIS based procedure "BIGBANG" [36,37] ("Nationwide GIS-based hydrological water budget on a regular grid"), which evaluated all the factors necessary for the monthly groundwater balance on the national scale. The four computed layers (two variables for each of the wildfire seasons, see Figures 7 and 8) were:

- Total Precipitation in winter and summer months, respectively; those layers represented the monthly cumulative precipitation (mm) from 1951 to 2019, averaged on the winter and summer wildfire season months, respectively.
- Mean Temperature in winter and summer months, respectively; those layers identify the mean average temperature (°C) from 1951 to 2019, averaged on the winter and summer wildfire season months, respectively.

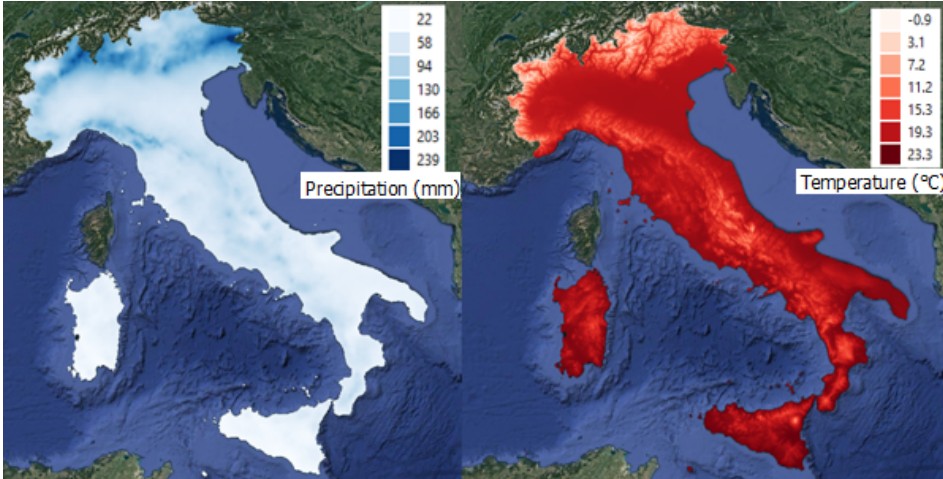

**Figure 7.** Layer of cumulative precipitation (mm) and temperature (°C) averaged in summer months (May–October) from ISPRA, *Istituto Superiore per la Protezione e la Ricerca Ambientale* (the retrieved time window is 1951–2019). The data were available at 1 km spatial resolution and was then re-sampled at the working project resolution (500 m) by the means of a bilinear interpolation algorithm.

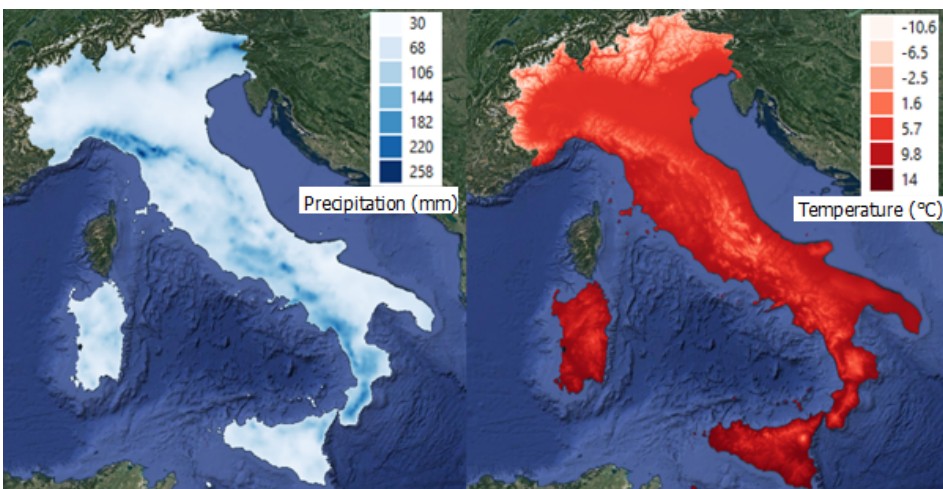

**Figure 8.** Layer of cumulative precipitation (mm) and temperature (°C) averaged in the winter months (November–April) from ISPRA (the retrieved time window is 1951–2019). The data were available at 1 km spatial resolution and was then re-sampled at the working project resolution (500 m) by means of a bilinear interpolation algorithm.

### 3.4.1. Vegetation Variables

The main data source used for computing vegetation cover variables is CORINE Land Cover 2018 (CLC2018) at the third level. [32,38]. The CLC2018 vector layer has been processed in the following way: removing all polygons corresponding to urbanized areas and non-burnable ares (e.g., water bodies), and then rasterizing the layer. The obtained raster, containing the CORINE code for the pixels, has been then processed in order to obtain the *neighboring vegetation variables* [25]. Those additional variables are used to associate with any pixel a resume of the vegetational surroundings, e.g., identifying homogeneity in vegetation, the interface between two main vegetation covers, or more complex patterns. For any pixels, a Moore neighborhood of order 2 (the 24 surrounding pixels around the analyzed one) has been considered. The frequency of appearance of the several vegetation types has been recorded. For instance, if a pixel is completely surrounded by CLC2018 code "313", that is, mixed forest, the variable "neighbouring_313" will be set to 1 and all the other "neighbouring_XXX" variables will be set to 0, for any other considered CLC2018 code. See Figure 9 for a graphical representation of the neighboring variables.

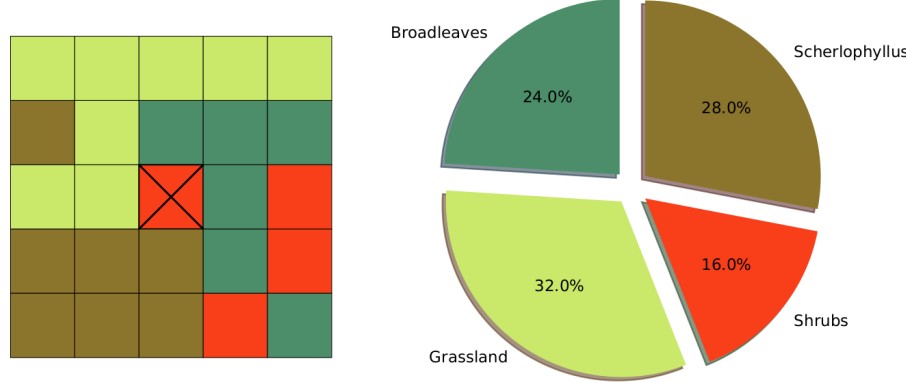

**Figure 9.** In this figure, a schematic representation of a Moore Neighborhood and the obtained "neighborhood percentages" of vegetation classes is shown. Of course, any other vegetation map not represented in the image is set to 0%.

Please note that the CLC18 map, however robust and reliable, at the third level accounts for only three kinds of forests: coniferous, broadleaves, and mixed forest. For

instance, it is thus incapable of distinguishing between Mediterranean conifers and firs. It is up to the ML algorithm to associate the third level CLC18 with the different climatic areas, height above the sea level and slope, etc., in order to explore all the possible configurations and enriching automatically the information contained in the broad vegetation categories.

### 3.4.2. Anthropic Variables

The following variables are connected to human activities and protected areas:

- the distance from the road network (m), obtained through processing of the SEDAC database [39];
- the distance from agricultural areas (m), obtained from the polygons of CLC18 land cover related to agricultural activities;
- the distance from settlements (m), obtained from CLC18 land cover polygons related to towns and settlements;
- the presence of Natura2000 protected areas (binary variable, 1 if the analyzed pixel falls into one of the areas reported by Natura2000 network, and 0 otherwise).

### 3.5. Methodology: The Machine Learning Model

The proposed methodology is based on a machine learning model, and structured as a classification problem. It uses a Random Forest Classifier (RF) [40] as an algorithm, in order to find a functional relation between the dependent variable (label) and the independent variables (predisposing factors), described in the previous paragraphs. RF is usually characterized by a good prediction accuracy and tolerance to noise and outliers in the input data. It has shown in literature good prediction ability in predicting spatial distribution of wildfire susceptibility [20,25]. The adopted RF model is composed of an ensemble of several classification trees, which are individually generated by bootstrap samples on the training data. RF overcomes the issue of overfitting on the input dataset that may affect a single classification tree. In RF, tree-specific bootstrap samples are obtained from the original input dataset, and an unpruned classification tree is grown for any of the latter samples. The final model has a probabilistic output, ranging between 0 and 1, which comes from the predicted probability of belonging to label "1", that is, to be associated with wildfire occurrence. Such probability is computed as the mean predicted class probability of the trees which compose the forest. The model has been implemented in *Python 3* through *scikit-learn* library [41]. The Random Forest is run with a set of 750 estimators (trees) and the limit of the square root of the total number of features to be used by each classification tree [25,42].

In this work, two ML models are trained, one with winter season wildfires and one with summer season wildfires, which ultimately lead to two distinct seasonal wildfire susceptibility maps, at 500 m resolution. In the training phase, the model is able to associate the geo-climatic characteristics with each spatial pixel to the presence of a wildfire event based on the past occurrences. The main data-set for each specific wildfire season is built by merging all the pixels with a label equal to 1, that is, pixels belonging to the merged burned polygons, with an equal number of randomly selected 0-labeled pixels (pseudo-absences, that is, pixels that did not experience any wildfire between 2007 and 2019). This has allowed for building balanced data-sets. Classification tasks carried out over balanced data sets are generally simpler to perform [43,44]. If all the pixels that never experienced a fire were to be considered, the algorithm would be biased towards predicting label 0, thus increasing the false negative rate. For the winter case, a total of 44,156 pixels have been considered, 22,078 of which proceeded from burned areas; As for the summer case, a total of 261,404 pixels have been considered, 130,702 of which are from burned areas.

These databases have then been split between the training ones (75 percent of the total entries) and the test ones (the remaining 25% of the database).

The training set has then undergone a spatial 4-fold cross validation [25] to assess the generalization capabilities of the model. If the analysis is only limited to a random sampling of input and outputs, training and test data may be characterized by *spatial autocorrelation*,

meaning that data samples close to each other hold similar characteristics. In order to see if the generalizing capabilities of the model are good enough, and the model is not overfitting the data, training and validation data can be selected far enough apart in the geographic space. In this work, the *spatial k-fold cross validation* with $k = 4$ folds is adopted. Methodologically, the k-fold cross validation consists of randomly dividing the data-set into k folds, holding out a fold at a time, training the model on the remaining k-1 folds and then validating the model using the fold left out.

The assessment of the validation phase is made by evaluating the average Receiving Operator Characteristics (ROC) Area under the Curve (AUC) for each fold. A value higher than 0.8 is generally considered a good value in order to consider the model general enough [45].

Once the Random Forest model is built, accessory information can be extracted from the trained classification trees. For instance, variable importance can be computed, in order to rank input factors by their relevance. The method is based on mean decrease in Gini impurity [40,46], provided by the routines of the *scikit-learn* Python library [41].

### 3.5.1. Model Testing

As already described, 25% of the data-set have been excluded from the training phase for both winter and summer seasons. Those pixels are used to analyze the performances of the ML models in the following ways:

1.  The Mean Squared Error (MSE). This performance indicator is evaluated as follows:

$$\frac{1}{n} \sum_{i=1}^{n} (Y_i - \hat{Y}_i)^2 \tag{1}$$

    where $n$ in the number of test pixels, $Y_i$ represents the true label information (in a discrete fashion, with 1 standing for burned pixel and 0 for a non-burned one) and $\hat{Y}_i$ represents the label predicted by the continuous (probabilistic) output of the ML model.

2.  ROC curves are computed using the prediction of the model on the test pixels. The related AUC is retrieved.

3.  The model accuracy over the testing data-set has been computed. It is here recalled that the overall accuracy of a binary classification model is defined as

$$\text{Acc.} = \frac{TP + TN}{TP + TN + FP + FN} \tag{2}$$

    where $TP$, $TN$, $FP$, and $FN$ stand for the four entries of the confusion matrix: True Positives, True Negatives, False Positives, False Negatives, respectively. The accuracy is the ratio between the correctly identified test data entries and the total size of the test set.

4.  The susceptibility map values have then been divided in groups, for both seasons, following predefined quantile ranges, which are shown in Table 2 [25]. The test pixels associated with past wildfires are then assigned to each of the classes and their distribution can be visualized in a histogram. If the susceptibility map is well built, most of the testing wildfire pixels should fall on the highest susceptibility classes.

**Table 2.** Susceptibility classes based on quantile ranges.

| Susceptibility Class | Quantiles Range |
| --- | --- |
| Very Low | 0–0.30 |
| Low | 0.30–0.50 |
| Medium | 0.50–0.80 |
| High | 0.80–0.95 |
| Very High | 0.95–1 |

### 3.5.2. Workflow

The overall process is resumed in the scheme below:

1. Gather the input layers for the predisposing factors, process them and align their spatial extent and projection; pre-process climate data season-wise.
2. Gather the shapefile data for wildfire occurrences, dividing it season-wise and rasterizing according to the working projection.
3. For each season, create an initial balanced database with random sampling for pseudo-absences (pixels not touched by wildfires).
4. Split the database into train and test sets.
5. For the training database, perform a 4-fold spatial cross validation building four different RF models and evaluating ROC AUC.
6. Build the RF model from the entire training set.
7. Compute performance indicators and variable importance ranking.
8. Evaluate the model for every pixel of the entire Study Area, in order to obtain the Susceptibility Map.
9. Compute quantiles of the Susceptibility Map and check the susceptibility distribution of the test burned pixels.

## 4. Results

### 4.1. Spatial Cross Validation

Figure 10 portrays the spatial subdivision of the study area during the 4-fold cross validation. Each block is a square with a side of approximately 75 km. For each fold, the training dataset is created using three folds, and the testing set is the one fold excluded. The performance in terms of ROC AUC is evaluated at each turn.

The AUC scores for the folds of the cross validation are reported in Table 3. For each wildfire season, the AUC for any of the considered spatial fold is computed. Such values range from a minimum of 0.80 to a maximum of 0.87 (obtained in the summer case).

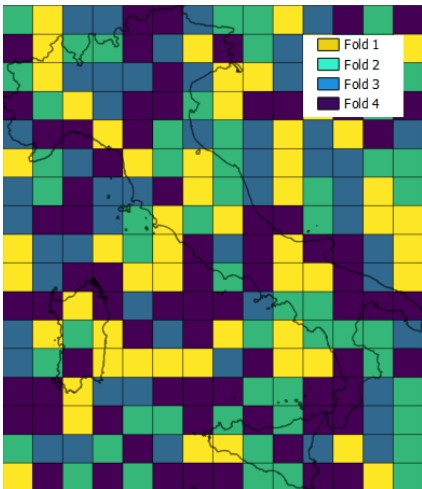

**Figure 10.** The spatial CV grid is represented. Each color corresponds to a different fold.

**Table 3.** AUC for the four folds of the spatial CV.

| Season | AUC Fold 1–4 |
|--------|--------------|
| Winter | [0.85 - 0.80 - 0.84 - 0.83] |
| Summer | [0.84 - 0.80 - 0.82 - 0.87] |

### 4.2. Input Features Ranking

The input predisposing factors (model features) are ranked by their relevance using the measure given by the Gini impurity. The splitting criterion used in RF is the Gini criterion,

used also in standard classification trees. At every split, one of the features is used to split the training data, which results in a decrease in the Gini impurity. The sum of all decreases in the forest due to a given variable, normalized by the number of trees, constitutes a measure for variable importance ranking [47,48]. The ranking of the predisposing factors for both ML models, the summer and the winter one, are reported in Table 4.

**Table 4.** Impurity-based importance measures for the Winter (left) and Summer (right) wildfire season

| Winter | Importance | Summer | Importance |
|---|---|---|---|
| Neighbour. Veg. | 0.30 | Neighbour. Veg. | 0.29 |
| Precipitation | 0.12 | Precipitation | 0.15 |
| Slope | 0.09 | Temperature | 0.11 |
| DEM | 0.08 | Slope | 0.08 |
| Temperature | 0.07 | DEM | 0.07 |
| Vegetation | 0.06 | Vegetation | 0.05 |
| North | 0.06 | Urban Dist. | 0.05 |
| Urban Dist | 0.05 | North | 0.05 |
| East | 0.04 | East | 0.04 |
| Roads Dist. | 0.04 | Roads Dist. | 0.04 |
| Crops Dist. | 0.02 | Crops Dist. | 0.02 |
| Natura 2000 | 0.01 | Natura 2000 | 0.01 |

Technically speaking, the importance of vegetation of each pixel and the neighboring vegetation reported in Table 4 are the sum of the variable importance of all the different vegetation classes; the vegetation categorical variable had been previously processed via the One Hot Encoding technique [49].

The values about neighbouring vegetation variables can be exploded in order to appreciate the contribution from every single vegetation cover to the relative score in mean decrease of Gini impurity. All the related histograms (for neighboring vegetation variable and local vegetation variable, for both winter and summer seasons) are available in the Supplementary Material. In Figure 11, the neighboring vegetation feature importance is shown. For the four highest classes, the distributions of summer susceptibility values over the pixel with the corresponding land cover class are shown in Figure 12.

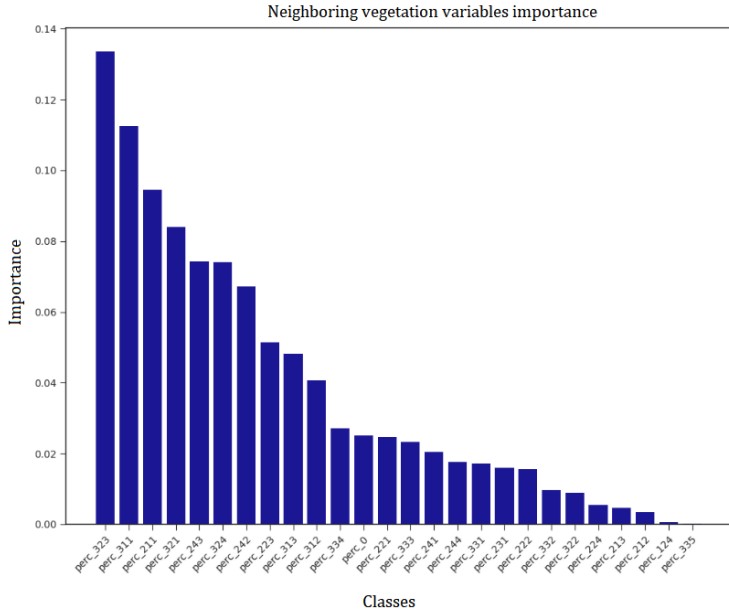

**Figure 11.** The neighboring vegetation feature importance for the summer season. The normalized output ranking is based on the mean decrease in Gini impurity associated with the ML model.

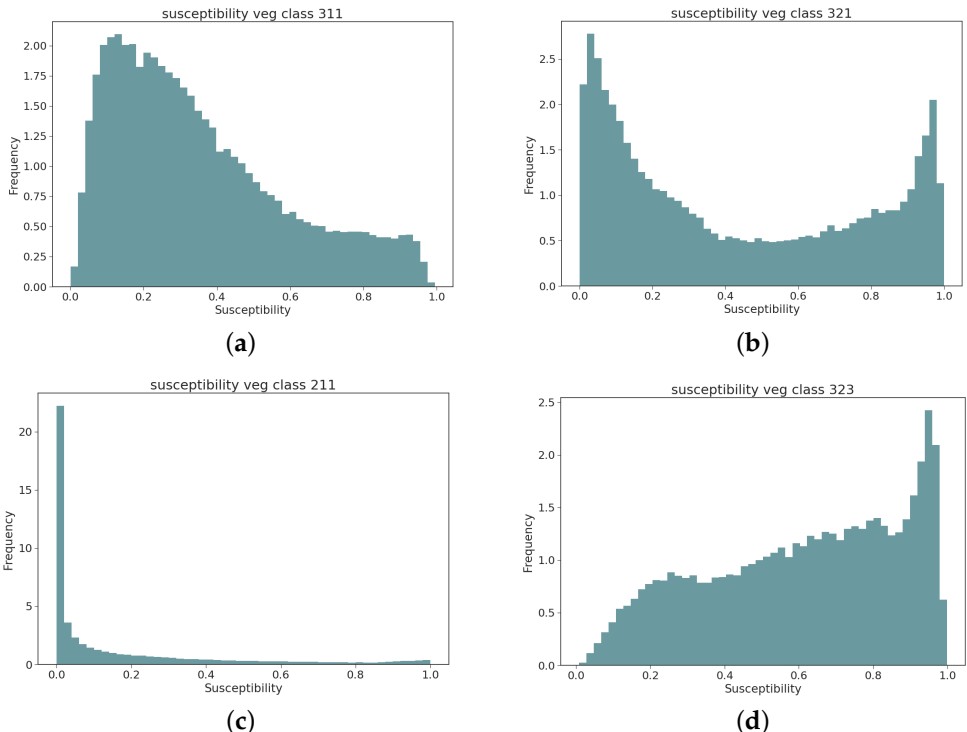

**Figure 12.** Susceptibility distribution of the four most important neighboring vegetation classes according to the feature rank of Figure 11. (**a**) Broad leaved, CLC code 311; (**b**) Natural grassland, CLC code 321; (**c**) Arable land, CLC code 211; (**d**) Sclerophyllous, CLC code 323.

### 4.3. Testing Phase: Performance Indicators

Different performance indicators have been assessed on the test set. More specifically, the ROC AUC (Figure 13 and Table 5), the mean squared error (MSE) and the accuracy of the model have been evaluated for both seasons. The first two indicators are assessed on the continuous (probabilistic) output of the model. On the other hand, the overall accuracy is calculated by the means of the binary output of the ML model. Such indicator is the ratio of all the test pixels correctly classified over all the test pixels. The discrete output of ML model also allowed for assessing how many data entries of the test set have been mis-classified, with the computation of the related confusion matrix (see Figure 14). The reported confusion matrices for winter and summer wildfire susceptibility models express similar features: there are generally more false positives than false negatives (19% in summer and 24% in winter versus 10% in both seasons) and less true negatives than true positives (81% in summer and 76% in winter versus 90% in both seasons).

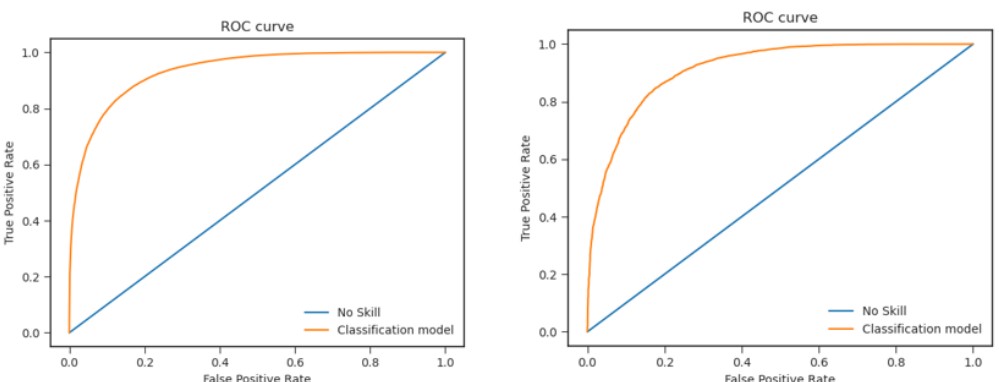

**Figure 13.** On the left, the ROC curve for the summer analysis. On the right, the ROC curve related to the winter season.

In addition, the seasonal MSE scores can be found in Table 5. The goal for a good ML model is having an MSE as close to zero as possible. Both seasonal scores are below 0.13.

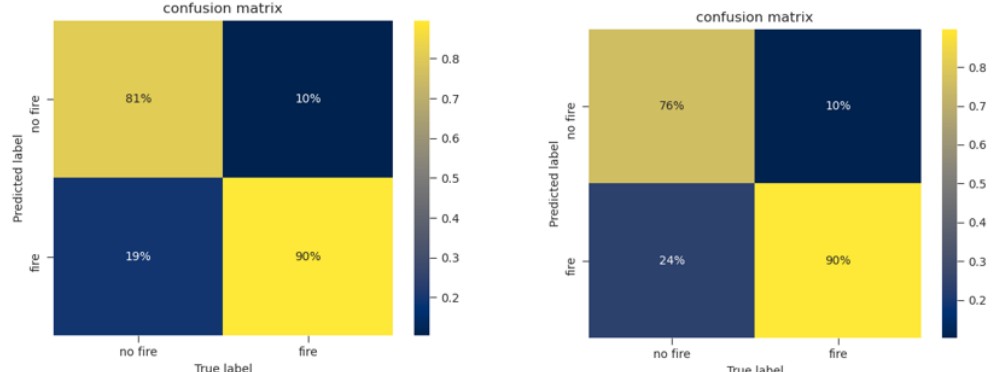

**Figure 14.** On the left, the confusion matrix for the summer analysis. On the right, the confusion matrix related to the winter season.

The overall accuracy of the model, divided between the two seasonal analyses, is presented in Table 5. The target value for this indicator is 1. In both seasonal models, the overall accuracy is above 0.8.

**Table 5.** ROC AUC, MSE, and overall accuracy performance indicators are listed for the Summer and Winter wildfire models, respectively.

| Season | ROC AUC | MSE | Overall Accuracy |
|--------|---------|-----|------------------|
| Summer | 0.93 | 0.107 | 0.85 |
| Winter | 0.91 | 0.122 | 0.83 |

As stated in Section 3, the produced susceptibility maps are divided into classes based on the value of selected percentiles [25] with the aim to partition the Study Area into homogeneous zones (from less to more fire prone) corresponding to different wildfire susceptibility levels (see Table 2).

The resulting seasonal maps are displayed in Figures 15 and 16.

*4.4. Quantiles Analysis: Distribution of Susceptibility Over the Test Set Burned Pixels*

As an additional performance indicator for the test database, it is the assessed the distribution of the susceptibility inside the burned pixels selected in the testing phase. The susceptibility is aggregated in the classes already presented for defining the color palette of the maps (see Table 2 and Figure 17).

Of course, the more test burned pixels lie on the upper 5th percentile of the susceptibility distribution, the more reliable is the map in identifying the fire prone areas. The results of this analysis are shown in Figure 17.

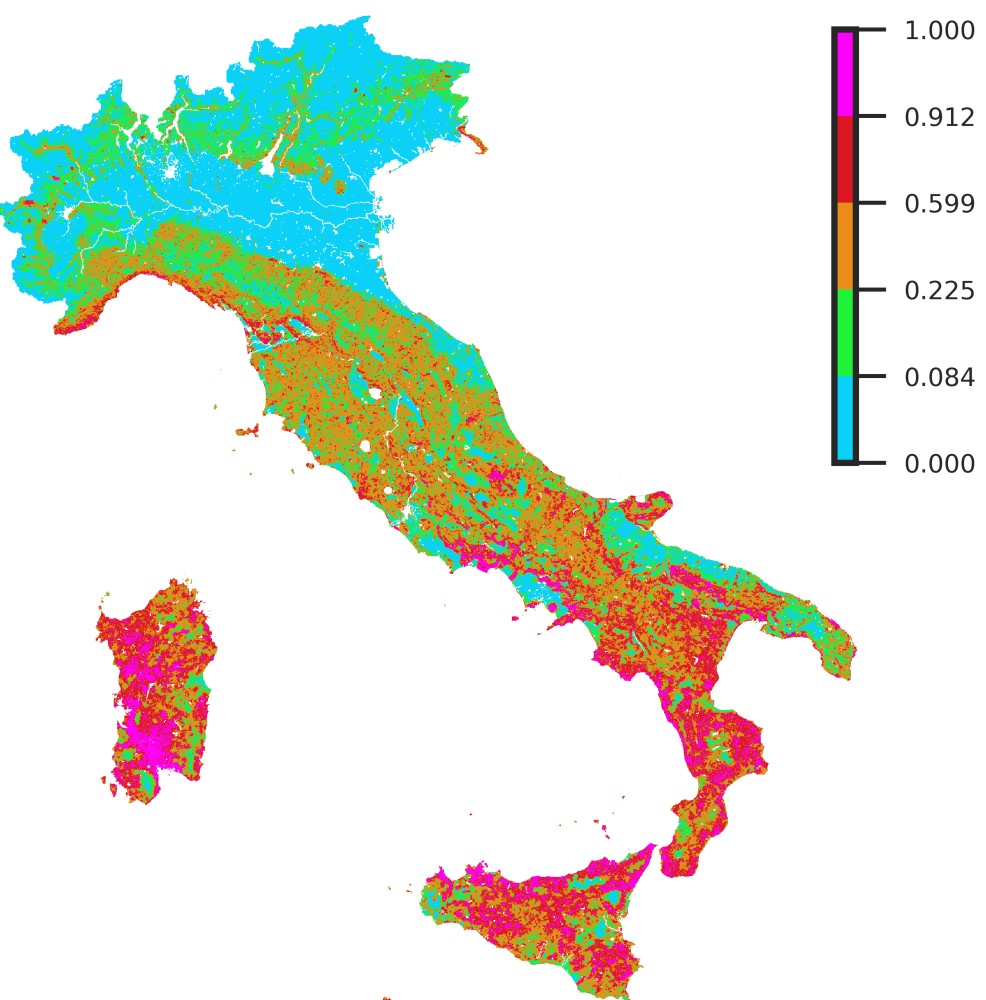

**Figure 15.** Susceptibility map for the summer fire season. To build the color palette, the continuous susceptibility values referred to the following percentiles have been calculated: 30–50–80–95. The color palette is therefore composed of five classes delimited by the susceptibility values of the following percentile class: blue (0–30), green (30–50), orange (50–80), red (80–95), purple (95–100). The computed numerical values are reported in the colorbar.

This straightforward analysis showed that more than 80% of the burned pixels of the test set fall into the highest susceptibility classes, which represent the 20 percent of the Study Area.

*4.5. Quantile Analysis: Distribution of Susceptibility Over a Set of Satellite-Retrieved Burned Areas from the 2021 Season*

In the following, another event-based validation process is performed. The distribution of the wildfire susceptibility over a selection of large wildfire events of Summer 2021 is analyzed. Such validation set, obtained from Sentinel satellite observations, is described in detail in Section 3.2. It is noted that those fires have never entered the ML process of the algorithm. This will thus constitute a useful benchmark in order to validate the produced summer map in terms of its predictive capability to identify future catastrophic events.

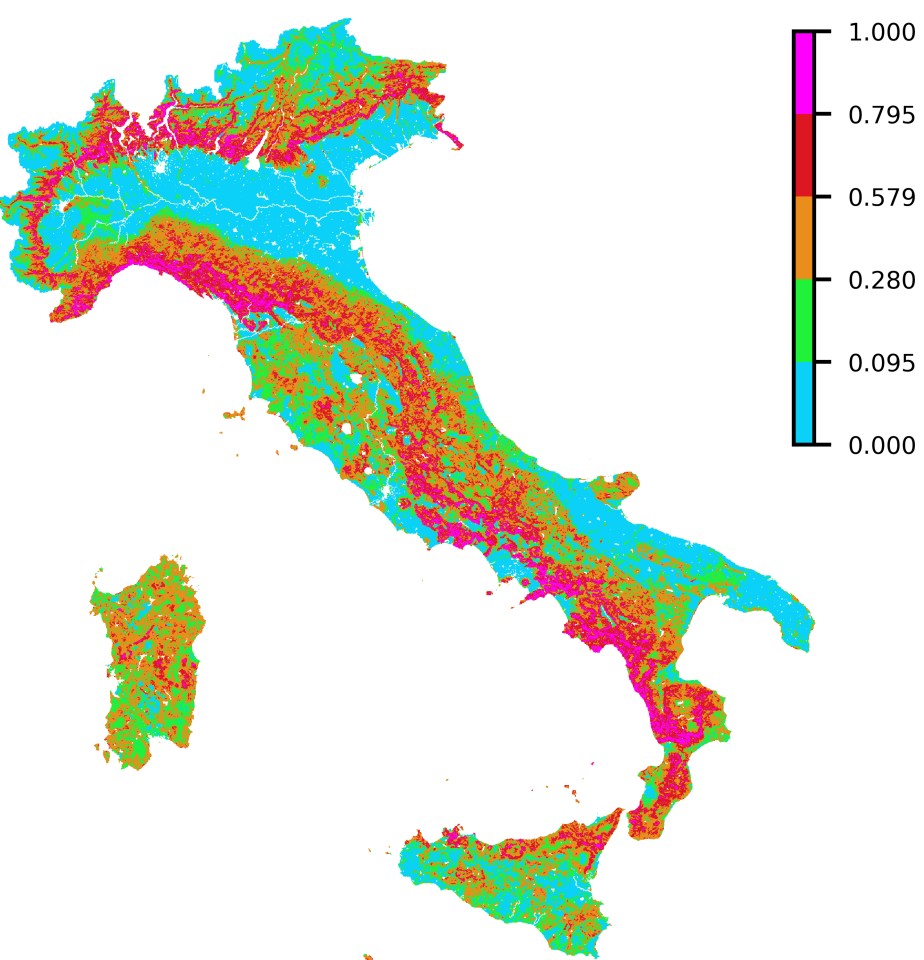

**Figure 16.** Susceptibility map for the winter fire season. To build the color palette, the continuous susceptibility values referred to the following percentiles have been calculated: 30–50–80–95. The color palette is therefore composed of five classes delimited by the susceptibility values of the following percentile class: blue (0–30), green (30–50), orange (50–80), red (80–95), purple (95–100). As for the summer case, the computed numerical values are reported in the colorbar.

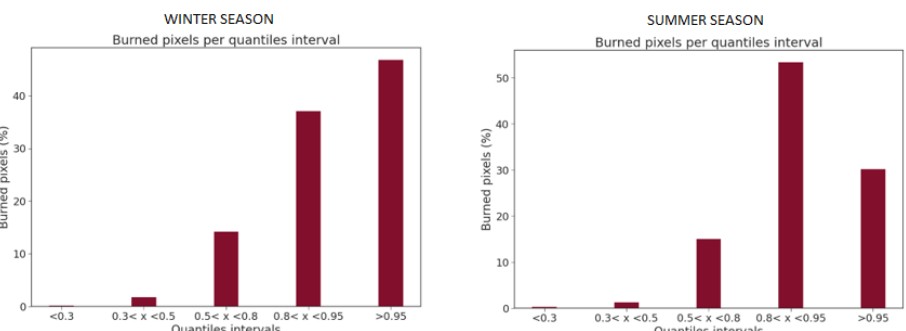

**Figure 17.** On the left, the susceptibility distribution grouped in percentile classes for the winter seasonal map. On the right, the susceptibility distribution grouped in percentile classes for the summer seasonal map.

The summer wildfire susceptibility map pixels corresponding to this wildfire data-set are assessed, and their values are divided into classes as per Table 2 and represented in Figure 18. The frequency distribution of susceptibility for the latter burned areas is also furnished. As expected, about 70% of Burned Area pixels fall into the two highest

classes (the red and violet classes in Figure 15), representing 20% percent of the Study Area. The susceptibility frequency distribution (right-hand side of Figure 18) confirms that results, with an increasing number of burned area pixels in the high susceptibility area. Figure 19 shows in detail two large fires of 2021 that affected the western part of Sardinia: the Montiferru wildfire (July 2021) of more than 11,000 hectares, and the Marghine fire (August 2021) that burned around 1400 hectares. After surrounding the Santu Lussurgiu municipality, the Montiferru wildfire rapidly moved northwestward, menacing the municipality of Cuglieri, causing hundreds of displacements among the affected population. The patterns exhibited by the wildfire susceptibility map over the latter wildfire are very interesting, since the hourglass-shaped burned scar follows the high susceptibility zones strictly, with a bottleneck in the middle. Concerning performance indicators, Montiferru and Marghine wildfires had an MSE of 0.102 and 0.007, respectively. The totality of Marghine wildfire spanned over pixels whose susceptibility was classified as High or Very High, while, in the Montifuerru wildfire case, about 80% of the related pixels fall into High and Very High susceptibility classes.

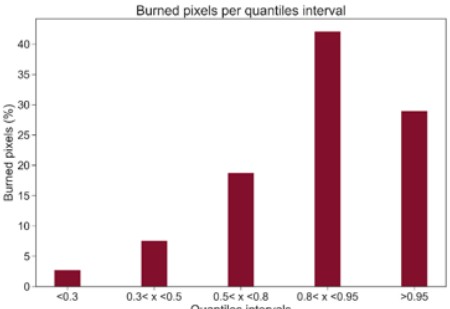
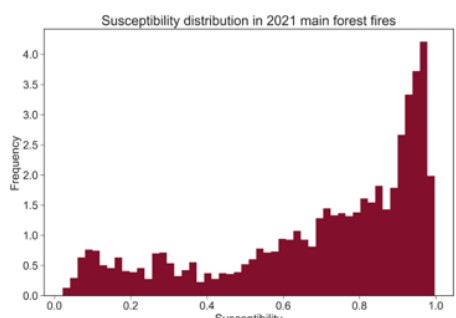

**Figure 18.** On the left, the 2021 burned area pixel distribution grouped by the five percentile classes (defined by the values of the following percentiles: 30, 50, 80, 95). On the right, the continuous distribution of the susceptibility inside the burned area related to the 2021 wildfires.

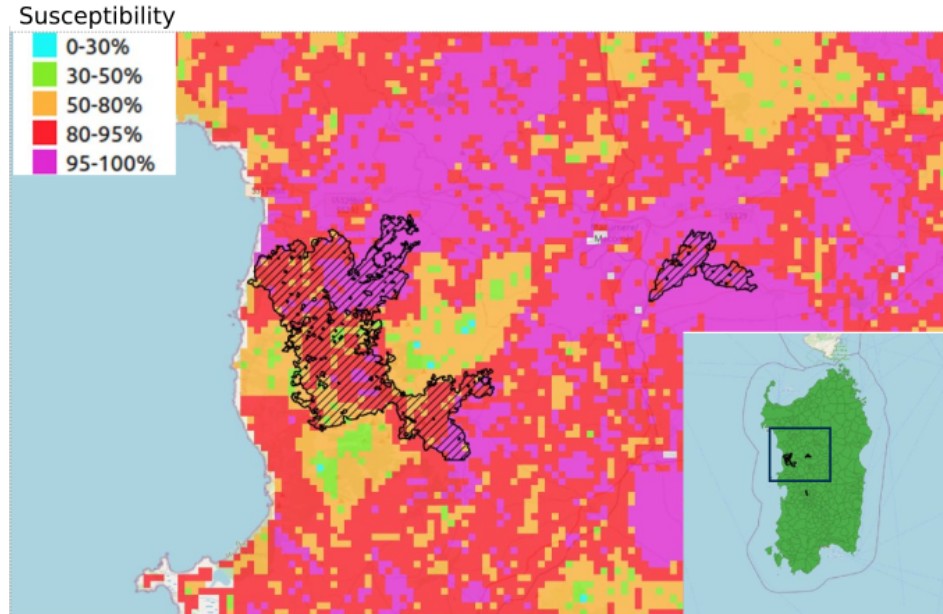

**Figure 19.** The North Western Sardinia island (see the bottom right box for the location of the examined wildfires in the island). The map portrays the spatial summer susceptibility produced by machine learning. The colors refer to the five classes of susceptibility (see Table 2). The satellite-retrieved burned scar of the Montiferru area wildfire (on the West, July 2021) and of Marghine (on the East, August 2021) are represented as hashed polygons.

## 5. Discussion

The proposed approach of assessing the wildfire susceptibility using an ensemble ML algorithm provides good results in terms of performance indicators. This is in line with the current literature in ML methods for evaluating the wildfire susceptibility, since ML for wildfire mapping has proven in general good accuracy and generality. However, most literature works focus on Study Areas at the regional scale [19,25,44,50] (with several notable exceptions of susceptibility maps at the national scale [21,24,51]). This has indeed motivated the presented work, in order to assess the solidity of the adopted framework. The spatial CV phase gave good results: of course, the final model, trained on the 75% of the available pixels, is characterized by better AUC, but satisfying results (AUC greater than 0.8 for any of the folds) have also been reached for the cross validation runs of the model, for both wildfire seasons. This means that the model, which is reaching high AUC values when run on the whole training dataset, is not lacking generality.

The produced susceptibility maps, similarly to what had been developed at the regional scale in [25], allow for assessing the zoning of wildfire prone areas for both winter and summer fire regimes. Typically, northern regions exhibit a winter fire regime [52,53], with particular focus on the Appeninic chain, and the alpine and pre-alpine areas. On the other hand, the southern region, also including the Italian islands of Sardinia and Sicily, is characterized by a summer wildfire regime.

The usefulness of the produced static maps lies in its ability to detect the areas where wildfire is more likely to occur in the future. The advances in this respect have been assessed by dividing the produced maps according to selected percentile intervals (because, of course, the probabilistic value given by the Random Forest prediction has not had any intrinsic physical meaning *per se*) and then makes use of tested burned pixels. This has been done in two ways. The first one used the randomly sampled burned pixels of the test data-set, which comes from a merging of the ground retrieved burned scars of past wildfires. In this case, the results were good, with more than 83% of the burned pixels assigned to the two highest susceptibility classes, for both seasons. Notably, in the winter case, the provided susceptibility map would allow in principle to concentrate fire fighting resources and prevention/prepredness activities in the five percent of the vegetated territory that would account for half of the total wildfire occurrences.

However, since the usefulness of a static map for wildfire management purposes has to be proven for the future, possibly catastrophic, events, the remote-sensing retrieved burned scars of the severe wildfire summer season of 2021 allowed a thorough testing of the produced maps. The results have shown a good prediction capability also in this more challenging case, with around 70% of the burned area of these wildfires belonging to the two highest classes of susceptibility. In particular, more than 30% of the satellite-retrieved burned area belongs to the top 5% percentile of the Italian summer susceptibility map.

However, this analysis was not only limited to the production of static maps since the built ML models allow for a variable importance in order to rank input factors by their relevance, as described in Section 3 and presented in Table 4. The method is based on a mean decrease in Gini impurity [40,46] provided by a function of the Python *scikit-learn* package [41].

Such feature importance ranking shows how the neighbouring vegetation plays a very important role in determining whether a pixel may experience a wildfire or not, whatever the wildfire season. The other important variables are related to the climate (precipitation and temperature), followed by the aspects' components (northing and easting), and by the anthropic factors (distances from urban areas, roads, and crops). The least relevant feature is represented by the binary information related to the presence of protected areas (Natura 2000 network).

The detailed importance of each vegetation type, for both wildfire seasons, and both for the single-pixel vegetation or the neighboring vegetation are represented in detail in the Supplementary Materials, while the neighboring vegetation importance for the summer season is portrayed in Figure 11. Those importance values are the Gini importances relative

to the single CLC code, before their aggregation in the list of Table 4. In addition, in the Supplementary Materials, the most important vegetation classes are examined in detail, for both summer (CLC codes 211, 321, 311, 323) and winter ( CLC codes 211, 311, 324, 242). Every pixel of the susceptibility map corresponding to each of latter CLC codes has been analyzed, and the distribution of the susceptibility values has been plotted. For the summer case, the four most important types of neighboring vegetation are represented in Figure 12. Those plots highlighted different behaviours of those important classes: some classes are important to the ML algorithm because they are immediately associated with low susceptibility, such as arable land (211), while others are important because they are strongly associated with high susceptibility output (such as Sclerophyllous/ maquis vegetation, class 323). Other classes exhibit more complex behaviour, such as broad-leaves (311) and natural grassland (321). In this case, the interactions with other predisposing factors, such as DEM, slope and climate, are needed by the ML algorithm in order to assign a susceptibility value to the pixels characterized by such vegetation types.

As previously mentioned, the CLC18 is considered here at the third level of detail, and many other used pieces of data come from open data-sets, except for the synoptic database of wildfire occurrences.

The good results achieved applying the described ML framework and demonstrate that, as long as the spatial perimeters of the considered wildfire events have a good level of accuracy, ML Techniques such as Random Forest can make use of quite general predisposing factors, combining them in order to explore all the possible configuration and interactions, overcoming the limitations that may originate from the broad classes of land use. In Figure 20, the different susceptibility distribution of the same CLC class (311, broadleaves) is shown for the northwestern part of Italy. Areas characterized by lower height above sea level and with vegetation heterogeneity (that is represented in the model by the neighbouring variables) exhibit higher susceptibility values when compared to the broadleaves located in the Alps and Apennines mountain ranges. These findings would thus motivate at any level the systematic and precise burned area retrieval that are of utmost importance in producing susceptibility and risk maps.

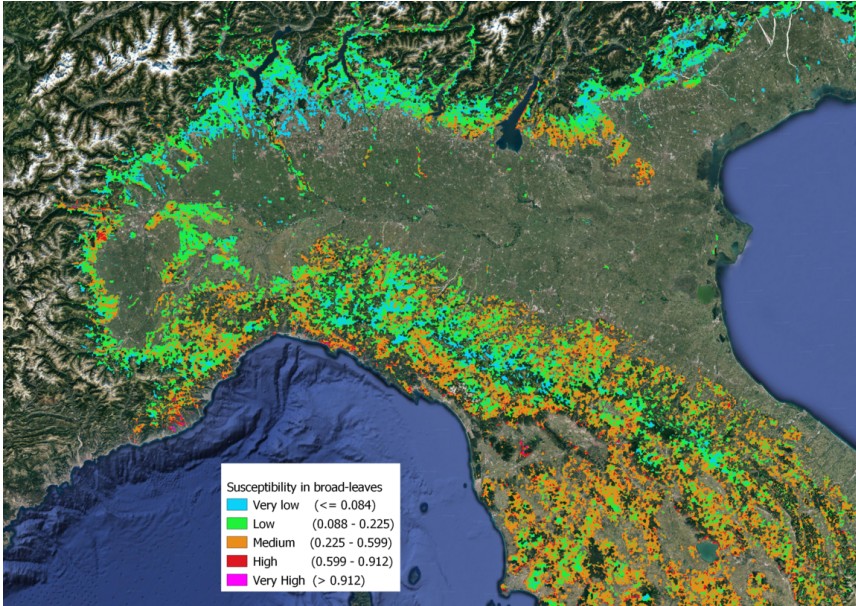

**Figure 20.** Details of the summer wildfire susceptibility over the pixels characterized by CLC18 class 311, corresponding to broadleaves, are represented, with the color palette of the national map of Figure 15. The figure refers to the broadleaves distribution over northwestern Italy. Even if the selected pixels share the same CLC land cover, there is a trend of higher wildfire susceptibility in close proximity to the coastline, and generally lower susceptibility in mountain areas, such as in the Alps and Apennines ranges.

## 6. Conclusions

In this work, a methodology to assess the wildfire susceptibility at the Italian national scale has been proposed. Two separate analyses have been performed for each of the wildfire regimes occurring in Italy, the summer and the winter one. The adopted model is based on the RF Classifier, and the problem is structured as a classification problem. The ML algorithm assigned to each pixel a value of susceptibility, after training on a balanced data-set based on the past wildfires' occurrences (label) and the pixels' geo-climatic and anthropic characteristics (features). RF being an ensemble model, it can return a probabilistic output, thanks to the contribution of different estimators (trees) to the classification task. The resulting classification on each pixel of the study area is associated with the wildfire susceptibility distribution. The proposed model gave satisfying results on the spatial CV and on the data-set, composed of randomly selected pixels, in terms of AUC, MSE and accuracy. A more operational oriented test has been performed on recent burned areas corresponding to the particularly severe 2021 summer season. In addition, for this benchmark, the performance remained good, confirming the good prediction capabilities of the adopted ML framework. Further analyses on the output of the model and on the input importance ranking showed how the ML is capable of obtaining good results relying on a rather simple description of the vegetation cover, when reliable wildfire polygons are provided along with the other explanatory variables.

Selected test cases give interesting food for thought on how to operationally use the produced maps that could potentially help with depicting the possible final extent of recently started wildfire events. Needless to say, event-specific studies should also take into account the dynamic effect of wind data, fuel moisture content, phenological state, and fire fighting actions on the fire front extent. Such contributions are not taken into account by the proposed static mapping, but are of course considered by the wildfire spread model and tools available in literature [54–57]. When the focus is on the single wildfire event, fires developed under particularly severe weather conditions or when fire fighting operations are compromised may also affect areas with low fire susceptibility [25].

Even if the proposed susceptibility maps can help Civil Protection Authorities and decision makers in wildfire management and long-term land use planning, they constitute the backbone of several possible hazard and risk mapping procedures, which may range from the static assessment to the dynamic one. Actually, analogous ML based susceptibility maps are used operationally by the Italian Civil Protection in order to modulate the outputs of dynamic forecasts of Fine Fuel Moisture Content, the potential rate of spread and fire-line intensity, embedded in the RISICO fire danger rating system [58–60]. The very same principle expressed in this work—the identification of reduced size areas where most of the wildfire events occur—is adopted in selected outputs of RISICO forecasts, in order to reduce overestimation of wildfire danger in low susceptibility areas.

The work presented in this paper constitutes a milestone of the modeling approach who started at the regional scale [25] and is now established at the national scale. Future works will be devoted to trans-boundary case studies, where susceptibility maps at the macro-regional scale could help in transboundary risk assessment procedures.

**Supplementary Materials:** Variable Importance charts are available online at https://www.mdpi.com/article/10.3390/fire5010030/s1: Figure S1: Distribution of the CORINE 2018 land cover classes in Italy, the class 0 refers to an aggregation of not burnable areas as specified in S1. Figure S2: Importance ranking (Mean Decrease in Impurity) of the neighboring vegetation variable related to the summer's seasonal analysis. Figure S3: Importance ranking (Mean Decrease in Impurity) of the winter's seasonal analysis. Figure S4: Importance ranking (Mean Decrease in Impurity) of the vegetation variable related to the summer's seasonal analysis. Figure S5: Importance ranking (Mean Decrease in Impurity) of the neighboring vegetation variable related to the winter's seasonal analysis. Figure S6: Normalized susceptibility distribution inside the non-irrigated arable land in the summer season (class 211). Figure S7: Normalized susceptibility distribution inside the broad leaved in the summer season (class 311). Figure S8: Normalized susceptibility distribution inside the natural grassland in the summer season (class 321).

Figure S9: Normalized susceptibility distribution inside the Sclerophyllous vegetation in the summer season (class 323). Figure S10: Normalized susceptibility distribution inside the non-irrigated arable land in the winter season (class 211). Figure S11: Normalized susceptibility distribution inside the rice fields in the winter season (class 242). Figure S12: Normalized susceptibility distribution inside the broad leaved in the winter season (class 311). Figure S13: Normalized susceptibility distribution inside the Transitional woodland in the winter season (class 211). Figure S14: The two histograms refer to the susceptibility distribution in the 2 wildfires presented in Figure 19: the Montiferru wildfire on the left hand side, and the Marghine wildfire on the right hand side. The susceptibility map presents values over 0.5 in almost the totality of the pixel affected by the two wildfires, with a good of values over 0.7. Figure S15. The two histograms refer to the susceptibility distribution in the two considered wildfires. The histogram on the left hand side refers to Montiferru wildfire, while the one on the right hand side to the Marghine wildfire. In both cases, most of the pixels fall on the two highest susceptibility classes. Table S1: CLC Codes and their description. The class evidenced by the (\*) symbol have not been used as input data of the machine learning model. For technical purposes, they have been merged into an *ad hoc* class "0".

**Author Contributions:** Conceptualization, A.T., G.M., P.F., A.G. and D.N.; Writing—original draft, A.T. and G.M.; Writing—review and editing, A.T., G.M., P.F., D.N., and A.G.; Investigation, A.T. and G.M.; Supervision, P.F., D.N. and A.G.; Software, A.T. and G.M.; Funding Acquisition, P.F. All authors have read and agreed to the published version of the manuscript.

**Funding:** This research was funded by the Italian Civil Protection Department—Presidency of the Council of Ministers through the convention with the CIMA Research Foundation.

**Acknowledgments:** The authors acknowledge the Italian Civil Protection Department—Presidency of the Council of Ministers, who funded this research through the convention with the CIMA Research Foundation, for the development of knowledge, methodologies, technologies, and training, useful for the implementation of national wildfire systems of monitoring, prevention, and surveillance. The authors would like to thank Blasi and Capotorti for kindly providing the Italian ecoregions' maps.

**Conflicts of Interest:** The authors declare no conflict of interest.

## Abbreviations

The following abbreviations are used in this manuscript:

| | |
|---|---|
| CLC | CORINE Land Cover |
| ML | Machine Learning |
| DEM | Digital Elevation Model |
| AUC | Area under the ROC curve |
| CV | Cross Validation |

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
