# Peer review of "Defining Wildfire Susceptibility Maps in Italy for Understanding Seasonal Wildfire Regimes at the National Level"

_fire, doi:10.3390/fire5010030_

Round 1

Reviewer 1 Report

The study uses random forest model to predict the susceptibility of a location to fire using historic fire perimeter data, climate data, and other ancillary datasets that are related to wildfire occurrence. I suggest the article is accepted after minor revisions. My specific comments and suggestions are attached.

Author Response

Please see attached PDF in order to see response to Reviewer #1.

Reviewer 2 Report

The paper under review is devoted to an important and urgent problem of forest fire forecasting. However, it is not entirely clear why this is being done at the national level. This would be much more useful at the local level, where specific actions are taken to prevent and combat forest fires.
The "probability" term is not appropriate in this research context because there is no mathematical justification for fire frequency, also the probability distribution law does not discussed. The authors themselves understand this and therefore use the vague term "susceptibility". However, in lines 4, 50, they unreasonably claim that susceptibility is the probability.
In the state-of-the-art overview, the authors indeservingly missed well-known fundamental probabilistic Schroeder's, Latham's models of fire ignition, and Rothermel's model of fire spread, as well as advanced geospatial applications based on these models already available (see https://www.firelab.org/applications).
Geospatial distributions of wildfire drivers in Fig. 3, Fig. 4 are historical and too coarse. The prediction of fire as a purely local phenomenon by them will be very unreliable. For accurate geospatial modelling of fire occurrence, the quasi-synchronous higher resolution remote sensed data are needed, for example, based on Copernicus Sentinel products. The vegetation types extracted from CLC18 do not reflect the natural fuel basic specifications and need a more detailed reclassification, at least taking into account the quantity (for instance by LAI) and condition (for instance by REP or TCI) of vegetation. Weather conditions were factored out. The distance from settlements, in my opinion, cannot be considered as a reliable feature of a fire threat: within a remote socially significant popular recreation zone, the fire occurence probability will be higher than in a city-close, but poorly visited place...
Thus, the input features for the ML model training, in my view, were taken unsuccessfully and, accordingly, the results obtained will be unreliable. In addition, the obtained maps of Fig. 12, Fig. 13 are low-detailed for practical purposes, and the 5 gradations of susceptibility used can hardly be called meaningful for the decision-makers. The analysis period of 3 months is also overlong.
Summing up, the reviewed research can be regarded as an abstract simulation of wildfire susceptibility, useful for ML models and algorithms debugging.
The authors mentioned the existing AutoBAM fire mapping system based on Sentinel-2 multispectral data, but it is not suitable for fire predicting. Moreover, the insufficient revisit frequency of Sentinel-2 (once every 6 days) significantly devalues the results of fire mapping.
I recommend the authors to refine this paper as follows: to revise the set of model input variables to take into account the fundamental physical drivers of fire occurrence, in particular - the local weather data; to compare the quantitative floating point values of fire susceptibility with the historical empirical frequency probability of fire occurrence at least inside a small-size local ground-truth site and to provide the resulting coefficient of determination.

Author Response

The response to Reviewer #2 area available in the attached PDF file.
Best Regards

The Authors

Round 2

Reviewer 2 Report

I think that the authors have done a sufficient revision of their paper and now it can be published.
In spite of that, this study in current form is only a theoretical demo of a possible approach to forest fire hazard assessment, which has very limited practical outcomes.
Unfortunately, the authors did not apply the vegetation condition assessment by multispectral satellite data, but still used the unified CLC2018 classification.
I hope that in future works the authors will be able to rescale output maps downto the local level and will make a significant contribution to the fire hazard prediction with high spatial and temporal resolution.

Author Response

To whom it may concern,
Please see the attached document for the reply to  Reviewer #2.
Best regards,
The Authors
